# Appendiceal Signet Ring Cell Carcinoma: An Atypical Cause of Acute Appendicitis—A Case Study and Review of Current Knowledge

**DOI:** 10.3390/diagnostics13142359

**Published:** 2023-07-13

**Authors:** Branko Andjelkovic, Bojan Stojanovic, Milica Dimitrijevic Stojanovic, Bojan Milosevic, Aleksandar Cvetkovic, Marko Spasic, Stefan Jakovljevic, Danijela Cvetkovic, Bojana S. Stojanovic, Danijela Milosev, Minja Mitrovic, Vesna Stankovic

**Affiliations:** 1Department of General Surgery, University Clinical Center Kragujevac, 34000 Kragujevac, Serbia; br.andjelkovic1990@gmail.com (B.A.); bojan.stojanovic01@gmail.com (B.S.); drbojanzm@gmail.com (B.M.); draleksandarcvetkovic@gmail.com (A.C.); drmspasic@gmail.com (M.S.); stefan_jakov87@yahoo.com (S.J.); 2Department of Surgery, Faculty of Medical Sciences, University of Kragujevac, 34000 Kragujevac, Serbia; 3Department of Pathology, Faculty of Medical Sciences, University of Kragujevac, 34000 Kragujevac, Serbia; danijelamilosevkg@gmail.com (D.M.); wesna.stankovic@gmail.com (V.S.); 4Department of Genetics, Faculty of Medical Sciences, University of Kragujevac, 34000 Kragujevac, Serbia; c_danijela@yahoo.com; 5Department of Pathophysiology, Faculty of Medical Sciences, University of Kragujevac, 34000 Kragujevac, Serbia; bojana.stojanovic04@gmail.com; 6Department of Neurology, Faculty of Medical Sciences, University of Kragujevac, 34000 Kragujevac, Serbia; minjam034@gmail.com

**Keywords:** signet ring cell carcinoma, appendix, acute appendicitis, appendectomy, case report, literature review

## Abstract

Appendiceal signet ring cell carcinoma (ASRCC) is a rare and aggressive form of appendiceal cancer, often presenting with nonspecific symptoms that overlap with acute appendicitis. Early diagnosis and appropriate management are crucial for improving patient outcomes in these rare malignancies. This case report and literature review aims to raise awareness among clinicians about ASRCC of the appendix as a cause of acute appendicitis and highlight the importance of considering this diagnosis in patients with atypical presentations or unexpected histopathological findings. We present a 65-year-old female patient with ASRCC who underwent successful surgical treatment and remains disease-free at the one-year follow-up. It also highlights the necessity of early detection and appropriate treatment in order to improve patient outcomes. In addition, a comprehensive literature review is provided, discussing the clinical presentation, histopathological characteristics, potential pathogenesis, treatment options, and prognosis of ASRCC.

## 1. Introduction

Primary adenocarcinoma of the appendix is an uncommon malignancy, representing less than 0.5% of all gastrointestinal neoplasms [1]. Among primary appendiceal carcinomas, appendiceal signet ring cell carcinoma (ASRCC) is an even rarer subset, constituting only 4% of all cases [2]. ASRCC is characterized by malignant cells containing large intracytoplasmic mucin vacuoles that displace the nucleus to the periphery, giving the cell a signet ring appearance [3]. While ASRCC can originate from various organs, the stomach is the most common primary site, making ASRCC an exceedingly rare and aggressive form of appendiceal cancer [4].

The clinical presentation of ASRCC is often nonspecific, with symptoms overlapping with those of acute appendicitis, such as right lower abdominal pain [5]. This overlap makes diagnosing ASRCC challenging, as it is frequently unsuspected before surgery [6]. Furthermore, ASRCC is considered an aggressive cancer, with metastases to adjacent organs, lymph nodes, or the peritoneal cavity present at the time of diagnosis in 93% of cases [1]. Consequently, early diagnosis and appropriate management are crucial for improving patient outcomes in these rare and aggressive malignancies [7].

The purpose of this case report and literature review is to raise awareness among clinicians about ASRCC of the appendix as a cause of acute appendicitis and highlight the importance of considering this diagnosis in patients with atypical presentations or unexpected histopathological findings. Additionally, this report aims to summarize the current knowledge on ASRCC, focusing on its clinical presentation, histopathological characteristics, potential pathogenesis, treatment options, and prognosis.

## 2. Case Presentation

A 65-year-old female patient was presented to the emergency department with a 3-day history of a worsening right lower quadrant abdominal pain, accompanied by nausea, vomiting, and low-grade fever. The patient reported no changes in bowel habits, weight loss, or any previous episodes of similar abdominal pain. She had no significant past medical history, including no prior gastrointestinal surgeries, and her family history was unremarkable for any gastrointestinal malignancies. She was a nonsmoker and consumed alcohol occasionally.

Upon physical examination, the patient appeared in moderate distress. Her vital signs were as follows: temperature 38.2 °C, blood pressure 130/80 mmHg, heart rate 98 beats per minute, and respiratory rate 18 breaths per minute. An abdominal examination revealed tenderness and guarding in the right lower quadrant, with a positive Rovsing’s sign and rebound tenderness. Bowel sounds were present but diminished.

The laboratory tests showed leukocytosis (white blood cell count 14,000/µL) with a predominance of neutrophils and an elevated C-reactive protein level of 140 mg/L (normal range: <10 mg/L), indicating an inflammatory response. Urinalysis, liver function tests, and serum electrolytes were within normal limits. Abdominal ultrasonography revealed a thickened, noncompressible appendix measuring 11 mm in diameter, with surrounding inflammatory changes consistent with acute appendicitis. No mass or cystic lesions were identified, and the ovaries and uterus appeared normal. The decision was made to proceed with surgical intervention.

The patient received intravenous fluids and antibiotics and underwent an urgent laparoscopic appendectomy. Intraoperatively, the appendix appeared inflamed and edematous, without evidence of perforation or abscess formation. A meticulous exploration of the peritoneal cavity was performed, and no gross pathological lesions or notable abnormalities were identified. The excised appendix was sent for histopathological analysis.

The histopathological analysis yielded a diagnosis of appendiceal signet ring cell carcinoma (ASRCC) with an extensive infiltration of the appendiceal wall (Figure 1). Macroscopically, the appendix measured 90 mm, with a thickened and whitish-gray wall and a congested external surface. Microscopic examination revealed signet ring cells with intracellular mucin vacuoles displacing nuclei toward the periphery. The signet ring cells were arranged individually and in syncytial patterns, with extracellular mucin fields infiltrating the entire wall thickness and invading the serosal layer (Figure 1A,B). Immunohistochemical analysis showed tumor cells expressing CK20 and CDX2, but not CK7, synaptophysin, or chromogranin (Figure 1C,D). The resection margins were clear of tumor cells, and no lymphovascular invasion was observed. The tumor was classified as T3N0M0, indicating locally advanced but non-metastatic disease, according to The American Joint Committee on Cancer staging system for appendiceal carcinoma (refer to Table 1 for the classification details).

After the diagnosis of ASRCC, the patient underwent a comprehensive staging evaluation, which included a computed tomography (CT) scan of the chest, abdomen, and pelvis. The CT scan did not reveal any signs of metastatic disease or lymphadenopathy. The patient’s tumor markers, including carcinoembryonic antigen (CEA) and cancer antigen 19-9 (CA 19-9), were also found to be within normal limits. The case was then presented at a multidisciplinary tumor board, and considering the locally advanced stage of the disease, the decision was made to perform a right hemicolectomy.

The patient underwent a successful right hemicolectomy with regional lymphadenectomy, and the histopathological analysis of the resected specimen did not reveal any residual tumor or lymph node involvement. The patient’s postoperative course was uneventful, and she was discharged on the seventh postoperative day. A multidisciplinary tumor board recommended adjuvant therapy with intravenous 5-fluorouracil (5-FU)/leucovorin (LV) due to the aggressive nature of ASRCC, and the patient completed six cycles of therapy. The patient has been regularly followed up and remains disease-free at the one-year follow-up, with no evidence of recurrence or metastasis.

## 3. Discussion

Appendiceal adenocarcinoma is a rare and uncommon malignancy of the gastrointestinal tract that arises from the glandular tissues of the vermiform appendix [9]. Despite its rarity, this cancer can be aggressive and has a high malignant potential [10]. It is typically diagnosed incidentally, following an appendectomy for acute appendicitis, as was the case in the patient described in this report [11]. Therefore, it is essential to consider a broad range of differential diagnoses in patients presenting with symptoms of appendicitis, particularly when imaging studies reveal atypical findings. Early detection and appropriate treatment are crucial for improving patient outcomes, and clinicians should be aware of the different histological categories of appendiceal malignancies and their associated prognoses [12].

### 3.1. Signet Ring Cell Carcinoma: A Rare and Aggressive Form of Appendiceal Cancer

The Surveillance, Epidemiology, and End Results (SEER) program, under the National Cancer Institute, has identified five histological types of appendiceal malignancies. These include colonic-type adenocarcinoma, mucinous adenocarcinoma, goblet cell carcinoid/adenocarcinoid, malignant carcinoid, and signet ring cell carcinoma [13]. The least common among these, signet ring cell carcinoma, stands out for its aggressive clinical course and extremely poor prognosis.

Signet ring cell carcinoma is a rare and particularly virulent form of appendiceal cancer. It is distinguished by a unique cellular morphology, where the cells contain a large vacuole that shifts the nucleus to the cell’s periphery, giving the appearance of a signet ring. If more than 50% of the tumor cells are of this type, it signifies the malignancy’s potential for rapid progression and its aggressive nature [14]. Alarmingly, the 5-year survival rate for signet ring cell carcinoma is only 7%, making it one of the deadliest types of cancer [1]. The tumor frequently spreads to neighboring organs, which has led some experts to categorize primary signet ring cell carcinoma (ASRCC) as a separate entity from other appendiceal cancers [6]. This stark discrepancy between the cancer’s rarity and its aggressive nature, along with its bleak prognosis, emphasize the urgent need for swift and accurate diagnosis. It also accentuates the need to view ASRCC as a distinct and particularly dangerous type of appendiceal malignancy.

Diagnosing primary ASRCC early is difficult, as its symptoms are often non-specific and can resemble acute appendicitis [5]. The only definitive way to diagnose it is by conducting a histopathological examination of a surgically removed appendix [15]. Thus, it is essential for healthcare providers to consider ASRCC when examining patients with non-specific symptoms, especially those with a history of acute appendicitis or appendectomy [11]. Timely detection and appropriate treatment are paramount to improving outcomes in patients with this rare and hostile cancer.

### 3.2. Demographic Patterns of ASRCC: Age, Sex, and Racial Disparities

The demographic statistics associated with these malignancies also warrant attention. The average age of onset for mucinous adenocarcinoma is 60 years, whereas signet ring cell carcinoma typically appears around 62 years of age, displaying a strikingly skewed male-to-female ratio of 1:11. Additionally, signet ring cell carcinoma has a predilection for the Caucasian population, as compared to other racial groups [15]. These demographic variations underscore the significance of considering age, sex, and race when investigating and diagnosing appendiceal neoplasms.

### 3.3. Pathogenesis and Genetic Associations of ASRCC

The pathogenesis of signet ring cell carcinoma is believed to involve genetic mutations in pluripotent intestinal crypt epithelial stem cells, leading to the formation of mucin droplets and neuroendocrine secretory granules in the epithelial intestinal crypts [16]. Understanding the pathogenesis and underlying mechanisms is important for developing effective diagnostic and therapeutic strategies for this aggressive malignancy.

Signet ring cell carcinoma of the appendix, although primarily sporadic in its occurrence, has been occasionally linked with specific genetic conditions, most notably hereditary nonpolyposis colorectal cancer (HNPCC), also known as Lynch syndrome [17]. Lynch syndrome is a genetic disorder characterized by a high risk of various cancers, particularly colorectal cancer, due to mutations in DNA mismatch repair genes. This mutation can lead to an accumulation of errors in the DNA, increasing the likelihood of cancerous growths [18]. The potential association between Lynch syndrome and ASRCC suggests that genetic counseling and testing might be recommended for some patients, especially those with a family history suggestive of HNPCC. However, further research is needed to clarify the precise nature and significance of this relationship.

### 3.4. Challenges in the Clinical Diagnosis of ASRCC: Overlapping Symptoms and Delayed Recognition

Clinically, primary ASRCC is a rare malignancy that presents with non-specific symptoms, making diagnosis challenging and often delayed. Most patients present with symptoms of acute appendicitis, such as right lower quadrant abdominal pain, nausea, vomiting, and fever, but some may present with atypical symptoms, such as weight loss, changes in bowel habits, or symptoms related to metastatic disease [5,19]. Due to the rarity of this malignancy and the overlap of symptoms with acute appendicitis, diagnosis is often made following a histopathological examination of the resected appendix as seen in our case.

### 3.5. Imaging and Diagnostic Challenges in Preoperative Evaluation of Appendiceal Neoplasms: Distinguishing Malignant Lesions from Acute Appendicitis

An accurate preoperative diagnosis of appendiceal neoplasms is crucial for planning appropriate surgical management in patients presenting with symptoms of acute appendicitis [20]. However, the lack of specific imaging findings can make it challenging to differentiate neoplastic lesions from other causes of acute appendicitis. Appendiceal adenocarcinomas frequently present similarly to acute appendicitis, resulting in frequent misdiagnosis [21]. However, imaging studies such as computed tomography (CT) or magnetic resonance imaging (MRI) can be useful in suggesting the possibility of a malignant lesion. Imaging findings suggestive of a malignant lesion may include a non-homogeneous mass in the appendiceal area with enhancing wall, peri-appendicular fat stranding, and surrounding lymph nodes [22]. Conversely, findings such as localized ascites, appendicoliths, intramural air, and focal cecal apical thickening are highly specific for acute appendicitis [23]. Additionally, advanced age at presentation may indicate the possibility of an underlying neoplasm [24]. An accurate diagnosis of signet ring cell carcinoma is often challenging, as it tends to have already metastasized by the time of diagnosis compared to other primary appendiceal tumors. To achieve a precise diagnosis, a comprehensive approach is necessary, which includes familiarity with the clinical presentation, radiologic features, colonoscopy results, and tissue sampling. Overall, a meticulous evaluation of imaging findings and patient characteristics can aid in the accurate diagnosis of appendiceal neoplasms, leading to appropriate surgical management.

### 3.6. Immunohistochemical Staining in the Diagnosis of ASRCC

Appendiceal signet ring cell carcinoma is characterized by the presence of malignant cells with abundant intracytoplasmic mucin that displaces the nucleus to the periphery, resulting in the appearance of a signet ring [14,25]. Immunohistochemical staining can be useful in differentiating primary appendiceal ASRCC from metastatic ASRCC from other primary sites. Primary appendiceal ASRCC typically stains positively for cytokeratin-20 and CDX-2, while staining negatively for cytokeratin-7, as was observed in our case [26].

### 3.7. Poor Prognosis of ASRCC: High Prevalence of Distant Metastases and Low Survival Rates

Appendiceal signet ring cell carcinoma generally has a poor prognosis, particularly when the cancer metastasizes to other organs or tissues [27]. As McGory et al.’s study suggests, this cancer type exhibits a high prevalence of distant metastases (60%) and a low 5-year survival rate (18%) [28]. Even when excised in a localized state, signet ring carcinoma has a lower 5-year survival rate than other appendiceal cancer histological types [27]. 

### 3.8. Appendiceal Signet Ring Cell Carcinoma: Heterogeneous Presentation, Management, and Outcomes—Insights from Case Reports and Studies

The heterogeneity in the presentation, management, and outcomes of ASRCC is illuminated through various case reports and studies, as summarized in Table 2. This rare malignancy can be variably present and its successful management crucially depends on early detection and prompt treatment. Adjuvant chemotherapy can offer significant benefits to certain patients, underlining the importance of a multidisciplinary approach to treatment. Although some patients have enjoyed extended survival periods following appropriate surgical intervention and subsequent adjuvant chemotherapy, others have suffered due to delayed diagnosis and treatment. Such disparities in outcomes emphasize the necessity for rapid recognition, precise diagnosis, and personalized treatment strategies in patients afflicted with signet ring cell carcinoma of the appendix.

### 3.9. Individualized Management of ASRCC: Surgical Approaches, Adjuvant Therapy, and Controversies

The optimal management of ASRCC is still uncertain and requires an individualized approach based on the patient’s clinical and pathological characteristics [10]. The treatment approach often depends on the cancer stage, and retrospective series offer guidance. For mucosal lesions in the early stage, appendectomy is considered curative [33]. However, for advanced-stage tumors, a right hemicolectomy with lymph node dissection may be necessary. In cases of undifferentiated adenocarcinoma or a significant invasion of the sub-mucosa or lymphatic invasion, secondary right hemicolectomy with lymph node dissection should also be considered [11]. Surgical intervention should be guided by the depth of invasion, as studies suggest no significant difference in an overall five-year survival for mucosa-limited tumors between localized resection and extended surgery [34]. Conversely, extended surgery is associated with a better overall survival for tumors invading the mucosa [1].

Adjuvant chemotherapy may be considered for appendiceal carcinoma in cases of perforation or the invasion of surrounding structures, but its role in treatment remains controversial [19]. The National Comprehensive Cancer Network (NCCN) guidelines recommend managing appendiceal adenocarcinomas similarly to colon cancer, which includes considering adjuvant treatment with 5-fluorouracil (5-FU)-based chemotherapy [10,11,32]. In our case, the patient was prescribed adjuvant therapy with intravenous 5-fluorouracil (5-FU)/leucovorin (LV) due to the aggressive nature of ASRCC, and received a total of six cycles of adjuvant therapy. Studies have suggested that adjuvant chemotherapy, such as the CAPEOX regimen (capecitabine and oxaliplatin), may improve survival rates in patients with signet ring cell carcinoma [35]. The role of adjuvant radiotherapy in appendiceal adenocarcinoma, particularly in ASRCC, is uncertain. Systemic chemotherapy is a viable option for patients with metastatic disease, while cytoreductive surgery and hyperthermic intraperitoneal chemotherapy (CRS/HIPEC) are the standard treatments for peritoneal dissemination from appendiceal cancer [36,37]. However, the benefit of CRS/HIPEC in cases of peritoneal dissemination from ASRCC remains controversial [38].

To streamline the management of patients with primary signet ring cell carcinoma of the appendix, Figure 2 presents the algorithm for the management of ASRCC. This outline emphasizes early diagnosis, surgical intervention, and adjuvant therapy. It provides recommendations for diagnostic imaging, surgical resection, adjuvant chemotherapy, and addresses metastatic disease and peritoneal dissemination management, underlining the importance of a multidisciplinary approach in treating this rare and aggressive malignancy.

Appendiceal signet ring cell carcinoma is a rare and aggressive malignancy with a poor prognosis. Accurate diagnosis and appropriate management are crucial for improving patient outcomes. It is essential for clinicians to be familiar with the various aspects of this disease, such as demographics, clinical presentation, diagnostic challenges, and treatment options. Table 3 provides a concise overview of these key findings and considerations for appendiceal signet ring cell carcinoma. 

## 4. Conclusions

In conclusion, appendiceal signet ring cell carcinoma is a rare and aggressive malignancy that presents with non-specific symptoms, making early diagnosis challenging. Accurate diagnosis and appropriate management are crucial for improving patient outcomes, and a multidisciplinary approach is necessary for optimal treatment. The use of imaging studies and histopathological examination of the resected appendix are crucial for accurate diagnosis. Surgical intervention remains the primary treatment for appendiceal malignancies, while adjuvant chemotherapy may be considered in selected cases. Overall, clinicians should consider appendiceal signet ring cell carcinoma in the differential diagnosis of patients presenting with symptoms of acute appendicitis, particularly those with atypical symptoms or unexpected histopathological findings. A heightened awareness and early detection may improve patient outcomes for this rare and aggressive malignancy.

## Figures and Tables

**Figure 1 diagnostics-13-02359-f001:**
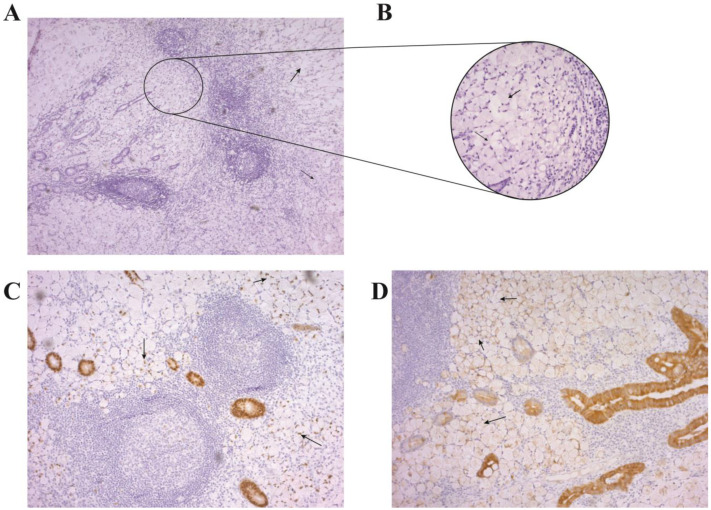
Histopathological examination of appendiceal signet ring cell carcinoma (ASRCC). (**A**) Hematoxylin and eosin (H&E) staining of the appendix showing signet ring cells and extracellular mucin (pointed out by black arrows) infiltrating the entire layer of the appendix (×50 magnification). (**B**) H&E staining of ASRCC. Black arrows indicate signet ring cells (×100 magnification). (**C**) Immunohistochemical analysis of ASRCC showing positive expression of CK20, indicated by black arrows (×50 magnification). (**D**) Immunohistochemical analysis of ASRCC showing positive expression of CDX2, indicated by black arrows (×50 magnification).

**Figure 2 diagnostics-13-02359-f002:**
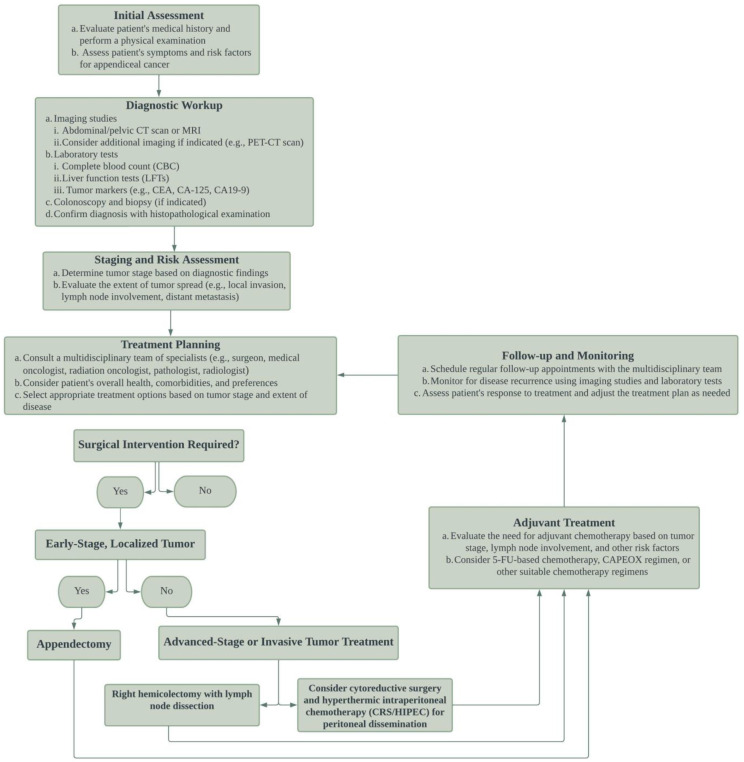
Algorithm for the management of appendiceal signet ring cell carcinoma (ASRCC). The figure outlines the diagnostic and treatment strategies for patients with primary ASRCC, including the importance of early diagnosis, surgical intervention, and adjuvant therapy. The algorithm includes recommendations for diagnostic imaging, surgical resection, and adjuvant chemotherapy, as well as considerations for the management of metastatic disease and peritoneal dissemination. The figure highlights the importance of a multidisciplinary approach to the management of this rare and aggressive malignancy.

**Table 1 diagnostics-13-02359-t001:** TNM staging system for appendiceal adenocarcinoma [8]. These adenocarcinomas, classified as mucinous or non-mucinous based on infiltrative invasion, are also graded by differentiation level. Mucinous neoplasms, showing pushing invasion without desmoplasia, are graded as low or high grade based on cytological atypia. These neoplasms can progress into pseudomyxoma peritonei, further classified into low, high, or high grade with signet ring cells (G1-3).

Stage	Classification		
**Primary Tumor (T)**			
TX	Primary tumor cannot be assessed.		
T0	No evidence of primary tumor.		
Tis	Carcinoma in situ refers to a condition where cancerous cells are present, but confined within the epithelial layer or the lamina propria. Specifically, the tumor remains restricted within the glandular basement membrane (intraepithelial) or within the lamina propria (intramucosal), and it does not extend through the muscularis mucosae into the submucosa. A new category, Tis (LAMN), has been introduced for low-grade appendiceal mucinous neoplasms. This category is used for those neoplasms that demonstrate a pushing margin but do not invade the muscularis propria.		
T1	Tumor invades the submucosa (through the muscularis mucosa but not into the muscularis propria).		
T2	Tumor invades the muscularis propria.		
T3	The tumor extends through the muscularis propria and invades into the subserosa or the mesoappendix. In this stage, acellular mucin or mucinous epithelium can be found within the subserosa.		
T4	The tumor breaches the visceral peritoneum, which may include the presence of mucinous peritoneal tumors within the right lower quadrant and/or direct invasion into other organs or structures. This stage also involves the presence of acellular mucin or mucinous epithelium in serosa.		
T4a	Tumor invades through the visceral peritoneum, including the acellular mucin or mucinous epithelium involving the serosa of the appendix or serosa of the mesoappendix.		
T4b	Tumor directly invades or adheres to adjacent organs or structures.		
**Regional lymph nodes (N)**			
NX	Regional lymph nodes cannot be assessed.		
N0	No regional lymph node metastasis (Note: regional lymph nodes include ileocolic nodes).		
N1	One to three regional lymph nodes are positive (tumor in lymph node measuring > 0.2 mm) or presence of tumor deposit(s) with negative lymph nodes.		
N1a	One regional lymph node is positive.		
N1b	Two or three regional lymph nodes are positive.		
N1c	No regional lymph nodes are positive but there are tumor deposits in the subserosa or mesentery.		
N2	Four or more regional lymph nodes are positive.		
**Distant metastasis (M)**			
M0	No distant metastasis.		
M1	Distant metastasis.		
M1a	Presence of intraperitoneal acellular mucin.		
M1b	Presence of intraperitoneal mucin with mucinous epithelium.		
M1b	Nonperitoneal metastases.		
Stage	T	N	M
Stage 0	Tis	N0	M0
Stage I	T1	N0	M0
	T2	N0	M0
Stage IIA	T3	N0	M0
Stage IIB	T4a	N0	M0
Stage IIC	T4b	N0	M0
Stage IIIA	T1	N1	M0
	T2	N1	M0
Stage IIIB	T3	N1	M0
	T4	N1	M0
Stage IIIC	any T	N2	M0
Stage IVA	any T	N0	M1a
Stage IVB	any T	N1	M1a
	any T	N2	M1a
Stage IVC	any T	any N	M1b

**Table 2 diagnostics-13-02359-t002:** Comparison of present case with previously published case reports on appendiceal signet ring cell carcinoma.

Case Report/Study	Age/Gender	Clinical Presentation	Treatment	Prognosis	Key Takeaways
Sato A et al., 2022 [6]	48/F *	Chronic abdominal pain, fullness, constipation, and diarrhea	Laparoscopic ileocecal resection, adjuvant chemotherapy (CAPOX)	Survived 2 years	Role of adjuvant chemotherapy in certain patients
Wang F et al., 2022 [29]	66/F	Abdominal distension and discomfort	Refused surgical treatment, adjuvant chemotherapy	Survived 5 months	Early detection and prompt treatment
Caesar-Peterson S et al., 2020 [16]	65/M *	Atypical abdominal pain	Appendectomy	N/A *	Early detection is crucial for improving outcomes
Vukovic J et al., 2018 [30]	22/M	Vomiting, diarrhea, and cramps in abdomen	Right-sided hemicolectomy and diverting ileostomy	Died after 2 months	Importance of early diagnosis and intervention
Kulkarni RV et al., 2015 [31]	45/F	Persistent right lower quadrant abdominal pain	Appendectomy and unilateral salphingo-oophorectomy	N/A	Multidisciplinary approach to management
Fusari M et al., 2012 [5]	80/M	Acute appendicitis	Appendectomy and subsequent right hemicolectomy	N/A	Consider appendiceal cancer in acute appendicitis to plan appropriate treatment
Suzuki J et al., 2009 [32]	66/F	Colonic obstruction and ovarian tumors	Hartmann’s operation, ileocecal resection, and bilateral salpingo-oophorectomy, refused postoperative chemotherapy	Survived 1 year	Importance of early recognition and appropriate surgical intervention
Ko YH et al., 2008 [15]	67/F	Abdominal distention due to unresectable peritoneal dissemination and ovarian metastases	Appendectomy and bilateral salpingo-oophorectomy, palliative systemic chemotherapy (FOLFOX-4)	Survived 1 year	Consider appendiceal cancer in unexplained ascites
Present case	65/F	Acute appendicitis	Appendectomy and subsequent right hemicolectomy, palliative systemic chemotherapy (5-fluorouracil (/leucovorin)	Survived 1 year	Accurate diagnosis and appropriate management are crucial for improving patient outcomes

* N/A, Not available; M, male; F, female.

**Table 3 diagnostics-13-02359-t003:** Summary of key findings and considerations for appendiceal signet ring cell carcinoma.

Category	Key Findings and Considerations
Prevalence	-Rare; represents 4% of all appendiceal neoplasms
Age and Sex Distribution	-Mean age of occurrence: 62 years; male-to-female ratio: 1:11
Clinical Presentation	-Symptoms often mimic acute appendicitis
	-Possible atypical symptoms: weight loss, changes in bowel habits
Histopathological Characteristics	-Malignant cells with abundant intracytoplasmic mucin
	-Nucleus displaced to the periphery (signet ring appearance)
	-Immunohistochemical staining: CK20+, CDX-2+, CK7-
Potential Pathogenesis	-Arising from genetic mutations in pluripotent intestinal crypt epithelial stem cells
Treatment Options	-Appendectomy for early stage disease
	-Right hemicolectomy with regional lymphadenectomy for advanced disease or high-risk features
	-Cytoreductive surgery with HIPEC for peritoneal dissemination
	-Controversial role of adjuvant chemotherapy
Prognostic Factors	-Advanced stage, lymph node involvement, lymphovascular/perineural invasion
	-Positive resection margins, peritoneal dissemination
	-Poor, with a 5-year survival rate of 7%; high prevalence of distant metastases (60%)
Diagnostic Challenges	-Nonspecific clinical presentation
	-Overlap with acute appendicitis symptoms
	-Lack of specific CT findings, tumor spread to adjacent organs
Management Strategies	-Early recognition and appropriate surgical intervention
	-Multidisciplinary approach to management

## Data Availability

As a case report and review article, no primary data are available for sharing. All data used in this article are referenced from previously published studies and patient medical records.

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
