# Peer review of "Appendiceal Signet Ring Cell Carcinoma: An Atypical Cause of Acute Appendicitis—A Case Study and Review of Current Knowledge"

_diagnostics, 2023, doi:10.3390/diagnostics13142359_

Round 1
Reviewer 1 Report
The manuscript by Andjeklovic, Stojanovic et al, is a case report of one patient and state of the art discussion on Appendiceal Signet Ring Cell Carcinoma. The manuscript looks well written and a good review on this rare cancer.
Comments:
1. Can the authors provide better images? The H and E and the IHC are not of acceptable standard. It is all orange. Is the brighfield too dim? In these images it is unclear what the authors want to show.
2. Discussion spelling is not correct. Please check all spellings and grammar.
Minor spell check required.
Author Response
Dear Reviewer,
We are sincerely grateful for your insightful comments and constructive criticism on our manuscript. Your feedback has greatly aided in improving the quality of our work.
In response to your observation regarding the quality of the images, we have replaced them with improved versions. We appreciate your keen eye for detail and agree that the previous images were not up to the standard. The newly included images are clearer and better demonstrate what we intended to present.
We have thoroughly checked and corrected any spelling or grammatical errors in the discussion section of the manuscript, as you suggested. Your remark was crucial in ensuring the professionalism and readability of our work.
Once again, we deeply appreciate your time and effort in reviewing our manuscript. We believe that with your valuable comments, we have been able to refine our work to a higher standard.
Best Regards,
Milica Dimitrijevic Stojanovic, MD, PhD
Reviewer 2 Report
Reviewer’s comments to Author:
1. In the literature review, is there any difference between the clinical symptoms and imaging examinations reported in this case and previous cases?
2. What is the difference between the clinical symptoms, physical examination and imaging examination of this case and the general acute appendicitis? It can be provided as a reference and reminder for our clinicians.
3. The resolution of the pathological diagrams presented in this case report is not clear enough. It is not easy to interpret. Can you provide H&E and IHC color pictures with clear resolution?
4. Reference 8: "The American Joint Committee on Cancer: The 7th Edition of the AJCC Cancer Staging Manual and the Future of TNM. TNM tumor staging should refer to the latest version of AJCC, 2018, 8th edition for staging.《AJCC Cancer Staging Manual, 8th Edition》.
Author Response
Dear Reviewer,
We sincerely appreciate your comprehensive review and insightful comments on our manuscript. Your feedback is indeed valuable in refining our work and improving its quality.
- Regarding your first comment, this case represents one of the most common clinical manifestations of Appendiceal Signet Ring Cell Carcinoma (ASRCC) where patients initially present with symptoms and imaging signs of acute appendicitis. The ASRCC diagnosis is often only established intraoperatively and confirmed by the pathology report. Our aim was to highlight this typical clinical scenario and present all aspects of this malignant disease, from pathogenesis and clinical manifestations to diagnosis and prognosis. In particular, we focus on treatment strategies, where for the first time in the literature, we presented a treatment algorithm as a quick reference guide for clinicians.
- The clinical symptoms, physical examination, and imaging findings in cases of ASRCC often mimic those of general acute appendicitis, which makes this malignancy difficult to diagnose preoperatively. We agree that highlighting these similarities could serve as a crucial reminder for clinicians, and we will ensure that this comparison is clearly outlined in the revised manuscript.
- We appreciate your feedback regarding the resolution of the pathological diagrams. In response to your comment, we have replaced them with high-resolution H&E and IHC color pictures that clearly illustrate the pathological features we aim to highlight.
- Thank you for pointing out the outdated reference. We have updated the TNM tumor staging reference to the latest version of AJCC, 2018, 8th edition as per your recommendation.
We are immensely grateful for the time and effort you have dedicated to reviewing our manuscript. Your feedback has been integral to the improvement of our work.
Best Regards,
Milica DImitrijevic Stojanovic, MD, PhD